# Degradation of Diazepam with Gamma Radiation, High Frequency Ultrasound and UV Radiation Intensified with $H_2O_2$ and Fenton Reagent

Michel Manduca Artiles [1], Susana Gómez González [2], María A. González Marín [1], Sarra Gaspard [3] and Ulises J. Jauregui Haza [4,*]

[1] Higher Institute of Technologies and Applied Sciences, University of Havana, Ave Salvador Allende # 1110 e/Infanta and Rancho Boyeros, A.P. 6163, Revolution Square, Havana 10400, Cuba; manduca@instec.cu (M.M.A.); magonzalez@instec.cu (M.A.G.M.)

[2] Pharmaceutical Laboratories Aica⁺, Ave 23 s/n e/268 y 270 San Agustín, La Lisa, Havana 17100, Cuba; susana1991@nauta.cu

[3] Laboratoire COVACHIM M2E, EA 3592, Université des Antilles, BP 250, CEDEX, Pointe-à-Pitre 97157, Guadeloupe; sarra.gaspard@univ-antilles.fr

[4] Instituto Tecnológico de Santo Domingo, Ave de los Próceres, Santo Domingo 10602, Dominican Republic

[*] Correspondence: ulises.jauregui@intec.edu.do

**Abstract:** A degradation study of diazepam (DZP) in aqueous media by gamma radiation, high frequency ultrasound, and UV radiation (artificial-solar), as well with each process intensified with oxidizing agents ($H_2O_2$ and Fenton reagent) was performed. The parameters that influence the degradation of diazepam such as potency and frequency, irradiation dose, pH and concentration of the oxidizing agents used were studied. Gamma radiation was performed in a $^{60}Co$ source irradiator; an 11 W lamp was used for artificial UV radiation, and sonification was performed at frequency values of 580 and 862 kHz with varying power values. In the radiolysis a 100% degradation was obtained at 2500 Gy. For the sonolysis, 28.3% degradation was achieved after 180 min at 862 kHz frequency and 30 W power. In artificial photolysis, a 38.2% degradation was obtained after 300 min of UV exposure. The intensification of each process with $H_2O_2$ increased the degradation of the drug. However, the best results were obtained by combining the processes with the Fenton reagent for optimum $H_2O_2$ and $Fe^{2+}$ concentrations, respectively, of 2.95 mmol $L^{-1}$ and of 0.06 mmol $L^{-1}$, achieving a 100% degradation in a shorter treatment time, with a dose value of 750 Gy in the case of gamma radiation thanks to increasing in the amount of free radicals in water. The optimized processes were evaluated in a real wastewater, with a total degradation at 10 min of reaction.

**Keywords:** advanced oxidation process; wastewater; diazepam; gamma radiation; high frequency ultrasound; UV radiation; Fenton reaction

## 1. Introduction

Contamination of surface water, groundwater, and wastewater has increased in recent years due to the presence of so-called "emerging pollutants", such as drugs and pesticides [1–5]. Many investigations report the inefficiency of conventional wastewater treatment plants for eliminating persistent pollutants, and as a result, the presence of contaminants in effluent from treatment plants, rivers, lakes, and to a lesser extent in groundwater [1]. Diazepam (DZP) is the most prescribed benzodiazepine for its hypnotic, tranquilizing, and anticonvulsive properties, with levels in water bodies varying from 10 ng $L^{-1}$ to 1 µg $L^{-1}$ [6–12]. The presence of benzodiazepines affects the ecosystems in different ways [13,14]. Due to benzodiazepines interaction with the GABAA receptor, they may affect the function of the nervous system of non-target species, such as aquatic organisms [13]. On the other hand, Subedi et al. showed that zebrafish (Danio rerio) larvae

exposed to the mixtures of psychotic drug residues, including the benzodiazepines, had affected immune system and gene expression [14].

Considering the high impact of pharmaceutical products, it is very important to remove them from the wastewater before discharge. Several researches carried out in recent years point to the use of advanced oxidation processes (AOPs) as innovative technologies for the elimination of persistent pollutants [6,15–21].

For the degradation of DZP in aqueous medium, some studies report the use of ozone [18], ultraviolet (UV) radiation, and its combination with oxidizing agents [6,22–24]. However, the most studied AOPs for drug degradation in water use gamma radiation and high frequency ultrasound intensified with $H_2O_2$ and Fenton reagent [25,26].

In the present work, we studied the degradation of DZP in a synthetic matrix by radical attack ($HO^\bullet$, $e^-$ (aq), $H^\bullet$, $HO_2^\bullet$) formed in the processes of radiolysis, sonolysis, and photolysis in aqueous media as well as its intensification with $H_2O_2$ and the Fenton reagent [26–28]. We hypothesize that the use of oxidizing agents will generate an additional amount of highly reactive $HO^\bullet$ radicals, allowing a greater efficiency in the degradation of the molecule. Different parameters were optimized in the degradation processes to increase the drug removal efficiency. These final conditions were applied in the analysis of samples of real residual water.

## 2. Materials and Methods

For the DZP degradation studies, the three AOPs were carried out separately at an initial concentration of 20 mg $L^{-1}$ DZP (Sigma Aldrich, St. Louis, MO, USA) in acidified distilled water, at pH values between 2 and 3, adjusted with a 10% solution of concentrated $H_2SO_4$ (95–97% purity; Merck, Darmstadt, Germany).

Acetonitrile (HPLC grade purchased from Sigma-Aldrich) and orthophosphoric acid (85% supplied by Merck) were used for the analysis by High Resolution Liquid Chromatography (HPLC). Hydrogen peroxide (35%) was obtained from Fluka. 99.5% potassium iodide, 99.5% purity heptahydrate iron sulfate, and 98% sodium sulfite were purchased from Merck.

For gamma irradiation of the samples, an ISOGamma-LLCo irradiator, equipped with a $^{60}$Co source was used. The irradiation doses used were 0.1, 0.25, 0.5, 0.75, 1.0, 2.5, and 5.0 kGy at a dose rate of 3.73 kGy $h^{-1}$. All experiments were performed at a temperature of $30 \pm 2$ °C. In the experiments carried out with the gamma irradiation process, the aqueous solutions of DZP were packed in 50 mL bottles with screw cap, which reached a thermal equilibrium at room temperature and atmospheric pressure.

A high-frequency ultrasound horn (Meinhardtultraschall Technik, Leipzig, Germany) with flat transducer was used for the studies of sonolysis. Frequencies values of 580 and 862 kHz, with varying power levels, were evaluated. The degradation was carried out in a 1.5 L glass reactor at a controlled temperature ($25 \pm 2$ °C) and constant stirring at 300 $m^{-1}$.

The laboratory-scale artificial photolysis was performed with a UV lamp of 254 nm and a power of 11 watts. The degradation was performed in a 500 mL beaker with irradiated area of 78.5 $cm^2$ and a constant stirring at 300 $m^{-1}$. The solar UV degradation scaling process was carried out in a 5 L open channel flat reactor of (irradiated area of 1600 $cm^2$) connected to a 20 L recirculation tank at the rate of 1 L $min^{-1}$.

For the processing of the liquid samples treated in the three processes, a Shimadzu High Resolution Liquid Chromatograph was used, with a LC-20AD double channel pump, an automated SIL-10AL injector, and a UV detector with SPD-M20A type arrangement. The column used was a Merck reverse phase with modified C18 (100 mm $\times$ 4.6 mm, 5 µm) spherical silica packed in an isocratic regime with 1 mL $min^{-1}$ flow, and a ratio of acetonitrile: water acidified to pH 2 with $H_3PO_4$: methanol of 40:40:20 $v/v$. The injection volume was 10 µL and the wavelength used was 230 nm.

The mineralization analysis of the samples was carried out using a Total Organic Carbon Analyzer Shimadzu (TOC-V CSN), equipped with an infrared detector of the non-dispersive type. The sample was injected at 50 µL and the combustion process was carried

out in a quartz tube at 680 °C with a platinum catalyst. An oxygen flow of entrainment of the vapors was used at a rate of 150 mL min$^{-1}$. The mean relative error in each result was less than 6% and 2% for TOC and HPLC, respectively.

The degradation and mineralization values of the experiments were calculated by Equations (1) and (2).

$$Degradation\ (\%) = \frac{C_i - C_f}{C_i} \times 100 \tag{1}$$

where $C_i$ is the initial concentration of diazepam and $C_f$ is the final concentration of diazepam at a point other than the initial value.

$$Mineralization\ (\%) = \frac{TOC_i - TOC_f}{TOC_i} \times 100 \tag{2}$$

where $TOC_i$ is the initial value of the total organic carbon and $TOC_f$ is the total organic carbon at the end of the reaction.

In the characterization of the radiolytic transformations of the solvent we used the concept of radiolytic performance (*G-value*), referred to the number of molecules, free radicals, ions, excited particles, among others, that form or decompose when the system absorbs 100 eV of energy from ionizing radiation [29,30]. This factor is calculated according to Equation (3).

$$G - value = \frac{RNa}{D(6.24 \times 10^{16})} \tag{3}$$

where $R$ is the change in concentration of diazepam (M), $Na$ is the Avogadro number, $D$ is the absorbed dose (Gy), and $6.24 \times 10^{16}$ is the conversion factor of Gy to eV/L. For the conversion of the *G-value* to mol J$^{-1}$ multiplies by $1.04 \times 10^{-7}$ [31].

## 3. Results and Discussion

### 3.1. Effect of Operational Conditions on the Degradation of DZP by Radiolysis, Sonolysis and Photolysis

Figure 1 shows the best conditions achieved in the process of degradation of DZP in aqueous solutions at an initial concentration of 20 mg L$^{-1}$ by sonolysis and radiolysis. In the sonolysis process, three power values were tested for each frequency value used. For the 580 kHz frequency, the electric power outputs were of 1.4, 8.7, and 21.8 Watt and for 862 kHz of 2.1, 10.4, and 30.6 Watt. In the photolytic degradation, a unidirectional 11 W lamp was used. Figure 1 shows the results of DZP degradation by gamma radiation. It is observed that at doses lower than 500 Gy, the degradation of DZP is very low. However, for doses higher than 2500 Gy, a 100% elimination of the drug is obtained.

For doses below 500 Gy the *G-value* is low. In this case, the amount of DZP that is degraded is less than 2% which may be associated with weak collisions between the molecule and the radicals involved in the degradation process.

The drug degradation increases with adsorbed dose. At 1000 Gy an 83.4% elimination of the DZP corresponding to a maximum *G-value* of 0.015 mol J$^{-1}$ is reached. At dose values above this point the *G-value* decays again since the degradation of the DZP reaches 100%, decreasing the probability that molecules are formed or destroyed in the system.

In the degradation of DZP by sonolysis a maximum at 180 min of 28.27% is obtained at values of 862 kHz frequency and 30.6 watts of power. This is explained due to the fact that at higher powers, cavitation bubbles are formed with high rupture energies which generate a greater amount of HO$^{\bullet}$ radicals, thus increasing the probability of interaction with the molecule and its degradation.

In the degradation of DZP by photolysis a maximum of 37.97% was reached at 300 min of irradiation. Although this degradation value is higher than that obtained by sonolysis, the exposure time of the molecule to the radiation increases by 66%.

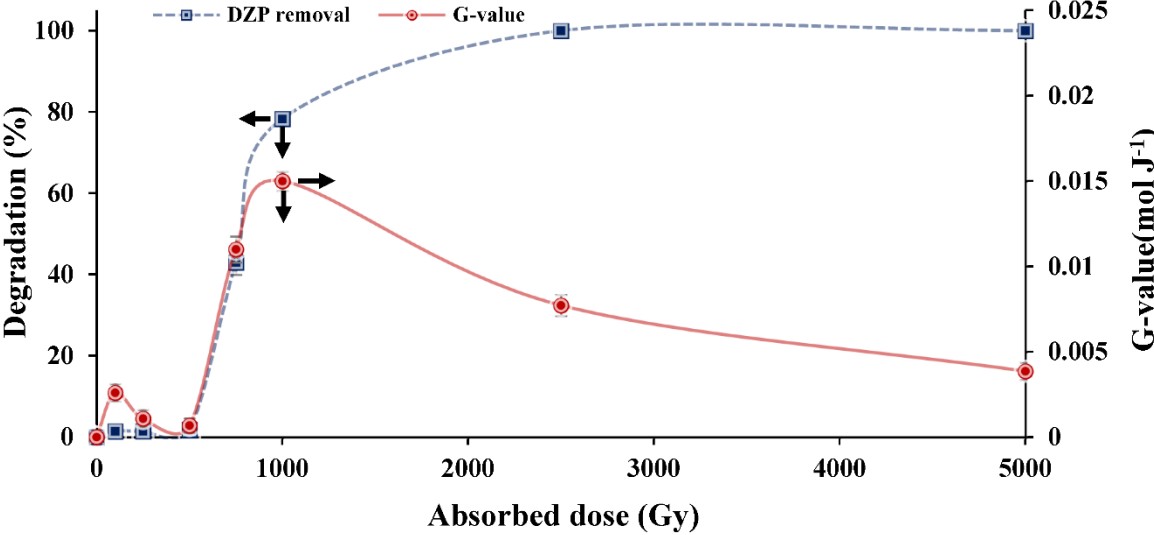

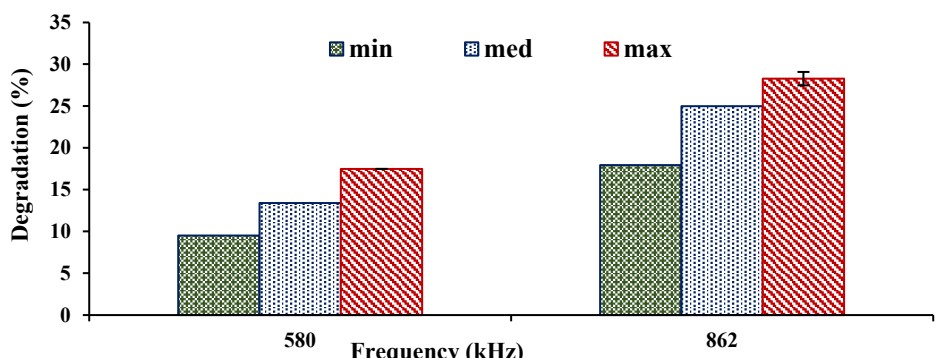

**Figure 1.** Degradation of DZP by gamma and utrasonic radiation. **Upper**: Influence of absorbed dose on DZP degradation and radiolytic yield (initial concentration: 20 mg L$^{-1}$, pH: 2.5, dose rate: 3.73 kGy h$^{-1}$); **Lower**: Influence of ultrasonic power and frequency on DZP degradation (initial concentration: 20 mg L$^{-1}$, pH: 2.5). Min, med, and max refer to the minimal, medium and maximal value of the ultrasonic power for each studied value of ultrasonic frequency.

### 3.2. Initial Effect of pH on Degradation of DZP

The initial pH is an important factor to study since it influences the chemical and physical conditions of the solution. The effect of pH on the radiolytic, photolytic, and sonolytic degradation of DZP (20 mg L$^{-1}$) at doses of 750 Gy, with a mercury lamp of 11 W, with a frequency of 862 kHz, and a power of 30.6 W.

The Figure 2 shows the influence of pH on DZP degradation. For radiolysis, drug degradation decreases with increasing pH, whereas for sonolysis at pH values 3, 5, and 7, degradation has a similar value (decreasing for pH 2.5 and 9).

Study of pH influence on sonolytic degradation shows that at pH 3, 5, and 7, the best degradation values are achieved. This result is closely related to the pKa = 3.4 value of the DZP corresponding to the carbonyl group present in its structure. This is explained by the fact that in acidic medium the molecule is dissociated in its protonated form, where the species with net positive charge, has an electrostatic interaction with the negative charges that are present in the periphery of the cavitation bubble [32], facilitating the process of drug degradation. In contrast, at pH less than 3, the degradation process is deprived, and hydrogen peroxide molecules formed (Equations (4)–(7)) can protonate and form more stable species such as $H_3O_2^+$, and or radicals HO$^\bullet$ can be attacked by the H$^+$ limiting the degradation process [32]. On the other hand, in basic medium (pH 9), there is practically

no presence of charged species in the dissociation equilibrium, so the interaction with the cavitation bubbles is much smaller [32]. Similar results have been obtained by other authors who report that the ultrasonic degradation of different compounds is higher in acid medium, which is also influenced by the fact that in this medium the HO$^\bullet$ radicals present a higher oxidation potential [33,34].

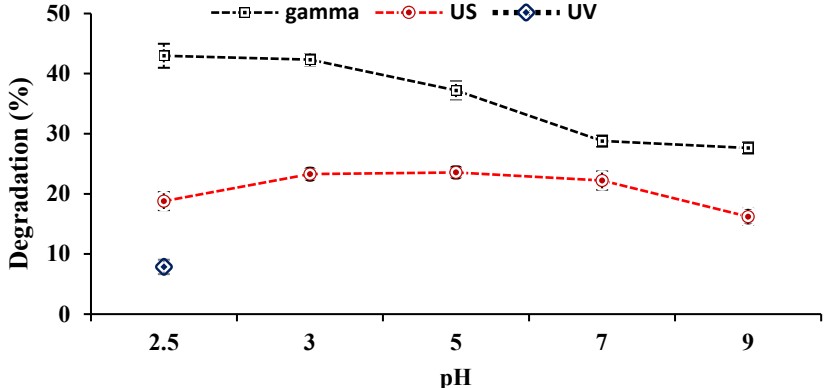

**Figure 2.** Influence of the pH of the solution on the sonolytic, radiolytic and photolytic degradation of the DZP (c (DZP) = 20 mg L$^{-1}$, T = 25 °C, US: v (agitation) = 300 m$^{-1}$; f = 862 kHz, t =180 min Gamma: D = 750 Gy, $\dot{D}$ = 3.73 kGy h$^{-1}$, UV: P = 11 W, λ = 254 nm, v (agitation) = 300 m$^{-1}$, t = 300 min).

$$H_2O + US \rightarrow H^\bullet + HO^\bullet \tag{4}$$

$$HO^\bullet + OH^- \rightarrow H_2O + O^- \qquad k = 1.2 \times 10^{10}\ M^{-1}s^{-1} \tag{5}$$

$$HO^\bullet + H_2O \rightarrow H_2O_2 + H^\bullet \tag{6}$$

$$HO^\bullet_{ac} + HO^\bullet_{ac} \rightarrow H_2O_{2\ (ac)} \qquad k = 5.5 \times 10^9\ M^{-1}s^{-1} \tag{7}$$

For radiolysis, increasing pH leads to a decrease in the degradation of DZP. On the other hand, in acidic medium the hydrated electron is more likely to react with the H$^+$ and form the radical H$^+$ by Equation (9) favoring the recombination reaction (Equation (8)). In the alkaline medium, the dissociation of the HO$^\bullet$ radical occurs, so degradation of the drug is not favored (Equations (8)–(10)). Different authors report the same behavior in the radiolytic degradation of other drugs with increasing pH [35–37].

$$H^\bullet + HO^\bullet \rightarrow H_2O \qquad k = 7.0 \times 10^{10}\ M^{-1}s^{-1} \tag{8}$$

$$e^-_{(ac)} + H_3O^+_{(ac)} \rightarrow H^\bullet + H_2O \qquad k = 2.3 \times 10^{10}\ M^{-1}s^{-1} \tag{9}$$

$$HO^\bullet \leftrightarrow H^+_{(ac)} + O^\bullet_{(ac)} \qquad pKa = 11.9 \tag{10}$$

In the case of UV radiation, the effect of the initial pH was studied only for the value 2.5. In a previous publication, this studied was carried out for a similar lamp to the one used here, showing that best degradation results are obtained for pH value of 2.5 [38].

*3.3. Effect of Hydrogen Peroxide on the Degradation of DZP by Sonolysis, Radiolysis and Photolysis*

In the combined process of photolysis and sonolysis with H$_2$O$_2$, an increase in the formation of the hydroxyl radical occurs (Equation (11)), which is the main responsible for the degradation of the molecule. By radiolysis the formation of HO$^\bullet$ radicals is intensified, due to the interaction of H$_2$O$_2$ with the solvated electron and the hydronium radical present in the water radiolysis (Equations (9) and (11)) [31,39].

$$H^\bullet + H_2O_2 \rightarrow HO^\bullet + H_2O \qquad k = 9.0 \times 10^7\ M^{-1}s^{-1} \tag{11}$$

Likewise, excess OH$^\bullet$ radicals in the medium can cause a decrease in the removal efficiency of the compound in the system. This is due to the fact that radicals tend to recombine at a very high reaction rate, as shown by Equations (7), (8) and (12)–(14) [31].

$$HO^\bullet + H_2O_2 \rightarrow HO_2^\bullet + H_2O \qquad k = 2.7 \times 10^8 \text{ M}^{-1}\text{s}^{-1} \qquad (12)$$

$$HO_2^\bullet + HO^\bullet \rightarrow H_2O + O_2 \qquad k = 6.0 \times 10^9 \text{ M}^{-1}\text{s}^{-1} \qquad (13)$$

$$HO^\bullet + e_{aq}^- \rightarrow OH^- \qquad k = 3.0 \times 10^{10} \text{ M}^{-1}\text{s}^{-1} \qquad (14)$$

For DZP degradation study shown in Figure 3, a 20 mg L$^{-1}$ solution of DZP at pH 2.5 for sonolysis, and pH 3 for photolysis and radiolysis was used. The power and working frequency was 30.6 W and 862 kHz, respectively, for the sonolysis. A 254 nm lamp with a power of 11 W was used for UV radiation, and the irradiation dose was 750 Gy for gamma radiation.

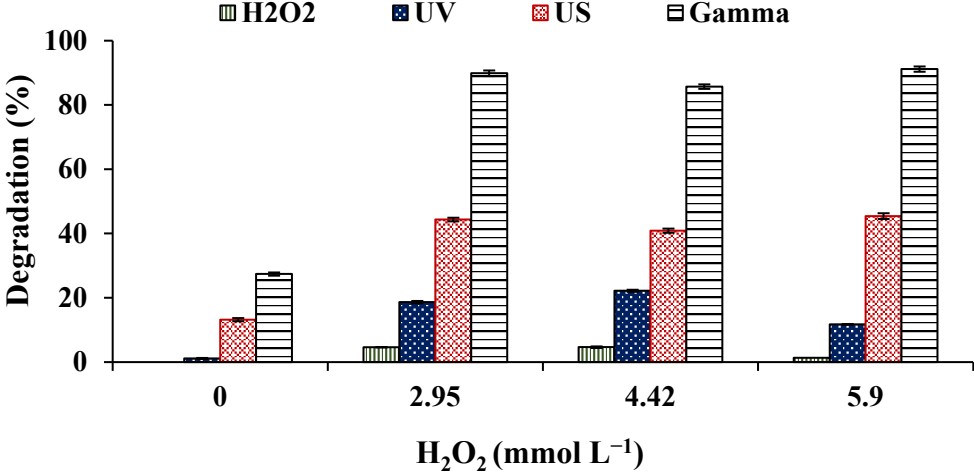

**Figure 3.** Influence of the addition of hydrogen peroxide on degradation of DZP (c (DZP) = 20 mg L$^{-1}$, T = 25 °C, t = 10 min US: v (agitation) = 300 m$^{-1}$; f= 862 kHz, pH = 3. Gamma: D = 750 Gy, $\dot{D}$ = 3.73 kGy h$^{-1}$, pH = 2.5, UV: v (agitation) = 300 m$^{-1}$, λ = 254 nm, P = 11 W, pH = 2.5).

In the three AOPs combined with H$_2$O$_2$ the DZP removal is intensified. For sonolysis and radiolysis, a maximum of 45.41% and 91.16% for a concentration of 5.9 mmol L$^{-1}$ of H$_2$O$_2$ is reached. For the photolysis the maximum removal of 22.2% is achieved at 2.95 mmol L$^{-1}$ of H$_2$O$_2$ decreasing for the higher concentration of oxidant used. This can be associated to processes of radical recombination, (Equations (3)–(5)) [21]. In general, the efficiency of the degradation process followed the order: AOPs with H$_2$O$_2$ > AOPs > H$_2$O$_2$.

The addition of H$_2$O$_2$ increases the efficiency of AOPs for the removal of contaminants in aqueous medium (You et al., 2021), obtaining the best results with the combination gamma/H$_2$O$_2$ as has been reported in other studies [31].

### 3.4. Effect of the Fenton Reagent on the Degradation of DZP by Sonolysis, Radiolysis and Photolysis

In Fenton processes, the HO$^\bullet$ radicals are generated by the catalytic decomposition of H$_2$O$_2$ using Fe$^{3+}$ ions in acid medium at pH in the 2–4 range [40]. This method facilitates a high formation of HO$^\bullet$ (Equation (15)), however an excess of Fe$^{2+}$ can trap them (Equation (15)), such as halogens, H$_2$O$_2$, or HO$_2^\bullet$ (Equation (16)) [31].

$$Fe^{2+} + H_2O_2 \rightarrow Fe^{3+} + OH^- + HO^\bullet \qquad k = 63 \text{ M}^{-1}\text{s}^{-1} \qquad (15)$$

$$HO^\bullet + HO_2^\bullet \rightarrow H_2O + O_2 \quad k = 1.0 \cdot 10^{10} \text{ M}^{-1}\text{s}^{-1} \qquad (16)$$

Table 1 shows the results of the degradation and mineralization of DZP by photolysis, sonolysis, and radiolysis combined with the Fenton reagent. Experiment 9 of each optimization series was repeated three times to verify the reproducibility of the experiments. In all three AOPs with the Fenton reagent, a 100% degradation is achieved for the $H_2O_2/Fe^{2+}$ ratio and concentration values of 5.9 mmol $L^{-1}$. The best experimental condition in which the maximum mineralization for the three AOPs is reached corresponds to the $H_2O_2/Fe^{2+}$ ratio of 29 with values of 4.42 mmol $L^{-1}$ and 0.15 mmol $L^{-1}$, of hydrogen peroxide and ferrous salt, respectively.

**Table 1.** Evaluation of degradation of diazepam intensified with Fenton reagent (c (DZP) = 20 mg $L^{-1}$, T = 25 °C, % D at 10 min, % M at 30 min US: v (agitation) = 300 $m^{-1}$ P = 30.6 W; f = 862 kHz; pH = 3. Gamma: D = 750 Gy; $\dot{D}$ = 3.73 kGy $h^{-1}$; pH = 2.5. UV: v (agitation) = 300 $m^{-1}$; λ = 254 nm; P = 11 W; pH = 2.5).

| Run | $H_2O_2$ (mmol $L^{-1}$) | $Fe^{2+}$ (mmol $L^{-1}$) | Fenton | | Sono-Fenton | | Gamma-Fenton | | Photo-Fenton | |
|---|---|---|---|---|---|---|---|---|---|---|
| | | | % D | % M | % D | % M | % D | % M | % D | % M |
| 1 | - | - | 13.2 | 27.4 | 1.1 | - | - | - | 13.2 | 27.4 |
| 2 | 5.90 | 0.59 | 98.0 | 7.3 | 100 | 14.3 | 100 | 37.5 | 100 | 10.6 |
| 3 | 5.90 | 0.20 | 78.2 | 4.3 | 80.8 | 17.9 | 94.9 | 51.2 | 86.7 | 14.8 |
| 4 | 5.90 | 0.12 | 71.1 | 2.8 | 74.8 | 17.4 | 92.8 | 50.1 | 46.5 | 14.5 |
| 5 | 2.95 | 0.29 | 89.6 | 1.9 | 100 | 14.4 | 96.1 | 48.7 | 79.9 | 14.0 |
| 6 | 2.95 | 0.10 | 73.2 | 2.1 | 74.3 | 19.4 | 96.3 | 58.4 | 64.3 | 16.9 |
| 7 | 2.95 | 0.06 | 51.3 | 3.0 | 58.7 | 18.3 | 96.4 | 56.7 | 35.8 | 16.4 |
| 8 | 4.42 | 0.44 | 97.2 | 9.6 | 100 | 15.0 | 96.5 | 53.0 | 98.1 | 15.2 |
| 9 | 4.42 | 0.15 | 71.6 ± 2.5 | 7.6 ± 0.8 | 65.9 ± 2.3 | 23.6 ± 1.1 | 95.3 ± 1.8 | 68.3 ± 2.7 | 47.7 ± 1.9 | 19.7 ± 0.7 |
| 10 | 4.42 | 0.09 | 69.1 | 4.9 | 59.0 | 22.5 | 95.3 | 66.0 | 37.0 | 19.3 |

The Fenton reagent alone can remove the DZP by 98% for the $H_2O_2/Fe^{2+}$ ratio value of 10 with concentrations of 5.9 mmol $L^{-1}$. In spite of their high percentage of elimination of the drug, only 9.6% of mineralization is obtained for the ratio 10 $H_2O_2/Fe^{2+}$ at concentrations of 4.44 mmol $L^{-1}$ and 0.44 mmol $L^{-1}$, respectively.

A multiple regression analysis yielded a mathematical model to evaluate the influence of $H_2O_2$ and $Fe^{2+}$ concentrations on DZP mineralization for each AOPs.

For the case of the US the percentage of mineralization responds to Equation (17) for a level of significance lower than 0.05 with a correlation coefficient of 0.89. The variable $(Fe^{2+})^2$ is not statistically significant and $H_2O_2$ concentration is proved to be the variable with the highest incidence in the mineralization process. Equation (18) represents the variation of mineralization for the photo-Fenton process.

$$\% M = -9.24 - 44.37\left[Fe^{2+}\right] + 15.92[H_2O_2] - 1.88[H_2O_2]^2 + 5.99\left[Fe^{2+}\right][H_2O_2] \quad (17)$$

$$\% M = -33.32 - 60.50\left[Fe^{2+}\right] + 49.12[H_2O_2] - 155.21\left[Fe^{2+}\right]^2 - 6.17[H_2O_2]^2 + 24.30\left[Fe^{2+}\right][H_2O_2] \quad (18)$$

where % M represents the percentage of mineralization, $[Fe^{2+}]$ the concentration of $Fe^{2+}$, and $[H_2O_2]$ the concentration of $H_2O_2$.

From the model the response surface and the isoline curves (Figure 4) of the mineralization were constructed as a function of $H_2O_2$ and $Fe^{2+}$ concentrations. The mineralization reaches a maximum higher than 22.5% for values ranging from 3.6 mmol $L^{-1}$ to 5.0 mmol $L^{-1}$ for $H_2O_2$, progressively decreasing with the increase of the catalyst. This may be due to the fact that the amount of $Fe^{2+}$ in the system begins to interact with the $HO^{\bullet}$ radicals, which decreases the process mineralization yield.

For the case of gamma radiation, the model adequately describes the experiment according to Equation (19) with a correlation coefficient of 0.99. All coefficients of the model are significant for a significance level of less than 0.05.

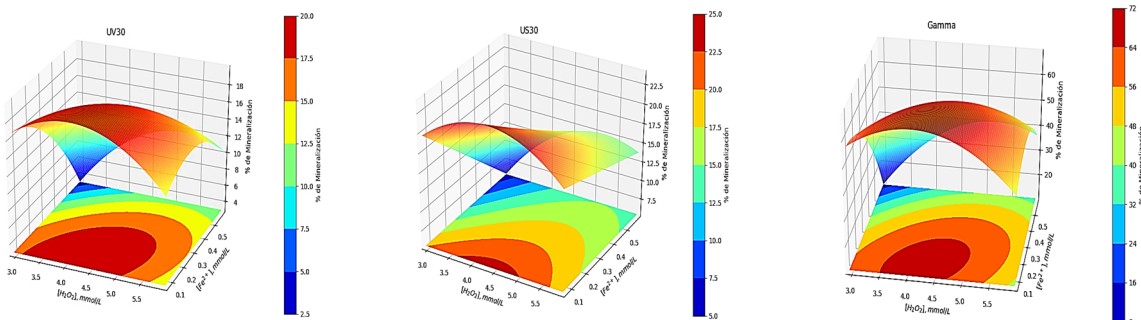

**Figure 4.** UV-Fenton, US-Fenton and gamma-Fenton processes. (DZP) = 20 mg L$^{-1}$, T = 25 °C, t = 30 min. v (agitation) = 300 m$^{-1}$, P = 30.6 W, f = 862 kHz, pH = 3. Gamma: D = 750 Gy, $\dot{D}$ = 3.73 kGy h$^{-1}$, pH = 2.5, v (agitation) = 300 m$^{-1}$, λ = 254 nm, P = 11 W, pH = 2.5.

$$\% \, M = -10.08 - 18.44\left[Fe^{2+}\right] + 14.49[H_2O_2] - 42.82\left[Fe^{2+}\right]^2 - 1.81[H_2O_2]^2 + 6.87\left[Fe^{2+}\right][H_2O_2] \tag{19}$$

Taking into account previous experiences using response surface methodology for optimizing AOPs [41], this procedure was used in this work. The response surface and the isoline curves were also constructed (Figure 4) where the mineralization reaches a maximum of 17.5% for H$_2$O$_2$ concentration values ranging from 3.3 mmol L$^{-1}$ to 5.0 mmol L$^{-1}$ and concentrations of Fe$^{2+}$ lower than 0.3 mmol L$^{-1}$ displaced towards an increase in the concentration of the former.

### 3.5. Evaluation of Energy Efficiency in the Degradation of DZP with AOPs

The estimation of energy consumption is an essential criterion for evaluating the most efficient AOPs in water treatment. A total of four criteria were used to estimate energy consumption in the degradation by radiolysis, photolysis, and DZP sonolysis. The first is the one proposed by Bolton and Carter in 1994, where the electric power is determined by order of magnitude (EE/O) (Equation (20)), which is expressed in kWh L$^{-1}$ and represents the energy required to degrade a liter of polluted with the drug water [30,42]. The second is the DW (Equation (21)) which expresses the amount of energy needed to degrade a milligram of pollutant and its unit of measure is kWh mg$^{-1}$ [31].

$$EE/O = \frac{Pt}{V log\left(\frac{C_o}{C_f}\right)} \tag{20}$$

where $P$ is the consumption power at the facility (kW), $t$ the irradiation time (h), $V$ the volume of the treated water (L), $C_o$ and $C_f$ the initial and final concentrations in the working solution (mg L$^{-1}$).

$$DW = \frac{Pt}{\left(C_o - C_f\right)V} \tag{21}$$

where $(C_o - C_f)V$ is the mass of the degraded contaminant (mg).

Energy efficiency in mineralization was evaluated using the criteria (*EE/O*) and c (*DW*) through Equations (22) and (23). The (*EE/O*) c represents the amount of energy per carbon needed to treat a Liter of contaminant and the (*DW*) c amount of carbon energy needed to degrade one milligram of pollutant.

$$EE/O = \frac{Pt}{V log\left(\frac{TOC_o}{COT_f}\right)} \tag{22}$$

$$DW = \frac{Pt}{\left(TOC_o - TOC_f\right)V} \tag{23}$$

where $P$ is the power of the equipment (kW), $t$ the process energy consumption time (h), $V$ is the effective volume (L), $TOC_o$ the initial total organic carbon, and $TOC_f$ the final total organic carbon.

In the study of energy efficiency, nine processes are compared in which three AOPs are used: radiolysis, photolysis, sonolysis, and their combination with oxidizing agents such as $H_2O_2$ and Fenton reactive. Experimental conditions were analyzed in which the best % degradation of DZP for each process and its combination with the oxidizing agents were reached.

The best results in the comparison of energy efficiency are found in the combination of AOPs with the Fenton reactive allowing to decrease irradiation times of 40 to 12 min for gamma-Fenton, 180 to 10 min in Fenton sleep and 300 to 10 min in photo-Fenton.

In Table 2, we report the best values obtained for energy efficiency for each studied process. The best results are obtained for the photo-Fenton reaction.

**Table 2.** Energy consumption in the degradation and mineralization (*) of DZP in sonolysis, photolysis and radiolysis. NA, not available.

| Process | Time (min) | EE/O (kWh L$^{-1}$) | DW (kWh mg$^{-1}$) | * (EE/O) (kWh L$^{-1}$) | * (DW) (kWh mg$^{-1}$) |
|---|---|---|---|---|---|
| Gamma | 40 | 1.2 | $2.9 \times 10^{-1}$ | NA | NA |
| Gamma-H$_2$O$_2$ | 12 | 1.7 | $9.7 \times 10^{-2}$ | NA | NA |
| Gamma-Fenton | 12 | $3.3 \times 10^{-1}$ | $8.8 \times 10^{-2}$ | 17.8 | 1.0 |
| Sonolysis | 180 | 2.5 | $6.5 \times 10^{-2}$ | NA | NA |
| Sono-H$_2$O$_2$ | 60 | $4.7 \times 10^{-1}$ | $1.3 \times 10^{-2}$ | NA | NA |
| Sono-Fenton | 10 | $3.9 \times 10^{-3}$ | $1.0 \times 10^{-3}$ | 0.5 | $2.0 \times 10^{-2}$ |
| Photolysis | 300 | 1.1 | $2.9 \times 10^{-2}$ | NA | NA |
| Foto-H$_2$O$_2$ | 300 | $1.3 \times 10^{-1}$ | $1.1 \times 10^{-2}$ | NA | NA |
| Foto-Fenton | 10 | $1.4 \times 10^{-3}$ | $3.7 \times 10^{-4}$ | 0.2 | $8.0 \times 10^{-3}$ |

*3.6. Study of the Photo-Fenton Process with Solar Radiation in a Real Wastewater*

Based on the positive results obtained in the experiments on the degradation of DZP at laboratory scale using the photo-Fenton process, it was decided to scale-up this process to a flat open-channel reactor with a reaction volume of 20 L. The experiments used sunlight as the energy source, combined with Fenton's reagent. The wastewater sample was taken at the entrance of the "María del Carmen" wastewater treatment plant in Havana city, Cuba.

Three degradation experiments of DZP were carried out at a concentration of 20 mg L$^{-1}$ for each molecule. The first experiment was carried out with technical water doped with the studied molecule, using the concentration of $H_2O_2$ and $Fe^{2+}$ ions that were the most efficient in the photo-Fenton process with artificial UV light. For DZP, an $H_2O_2$ concentration of 5.9 mmol L$^{-1}$ and an $Fe^{2+}$ ion concentration of 0.59 mmol L$^{-1}$ were used. The second experiment was carried out with wastewater doped with 20 mg L$^{-1}$ of diazepam and the same concentrations of $H_2O_2$ and $Fe^{2+}$ ions. In the third experiment, the COD of the wastewater sample was considered and the concentrations of $H_2O_2$ and $Fe^{2+}$ ions to be used were recalculated, they were 11.6 mmol L$^{-1}$ and 1.16 mmol L$^{-1}$ respectively. The actual wastewater was previously characterized as established in NC 27:2012, Table 3.

As can be seen in Figure 5, 10 min after the reaction started a total degradation was achieved for the molecule in the technical water. This behavior was similar to that obtained in the photo-Fenton process with artificial UV at the same concentrations of $H_2O_2$ and $Fe^{2+}$ ions. In the study, a total degradation of the DZP was obtained 30 min after the start of the reaction.

By increasing the concentration of the Fenton reagent, taking into account the COD of the wastewater, the total degradation of DZP was reached 10 min after the reaction started, a value that indicate the need of knowing the complexion of the matrix to improve the degradation process, and reaffirms the competitiveness of the organic molecules present in the wastewater for the HO$^\bullet$ radicals formed in the process.

**Table 3.** Analysis of wastewater before and after photo-Fenton treatment with sunlight.

| Parameters | Waste Water | Treated Water | Efficiency (%) | NC 27: 2012 |
|---|---|---|---|---|
| Temperature (°C) | 27 | 26 | - | <40 |
| pH | 7.65 | 7.32 | - | 6–9 |
| Conductivity ($\mu$S cm$^{-1}$) | 1085 ± 1 | 87 ± 1 | 91.9 | <2000 |
| CO (mg of $O_2$ L$^{-1}$) | 188 ± 30 | 61.4 ± 6 | 67.3 | <90 |
| BOD$_5$ (mg of $O_2$ L$^{-1}$) | 91 ± 1 | 36.5 ± 1 | 59.8 | <40 |
| Settleable solids (mL L$^{-1}$) | 2.5 ± 0.1 | 0 | 100 | <2 |
| Floating material | present | absent | - | - |
| Iron (mg L$^{-1}$) | 0.91 | 0 | 100 | - |
| TOC (mg de C L$^{-1}$) | 86.4 | 32.9 * / 23.7 ** | 61.9 / 72.6 | - |

\* TOC value at 30 min; \*\* TOC value at 120 min.

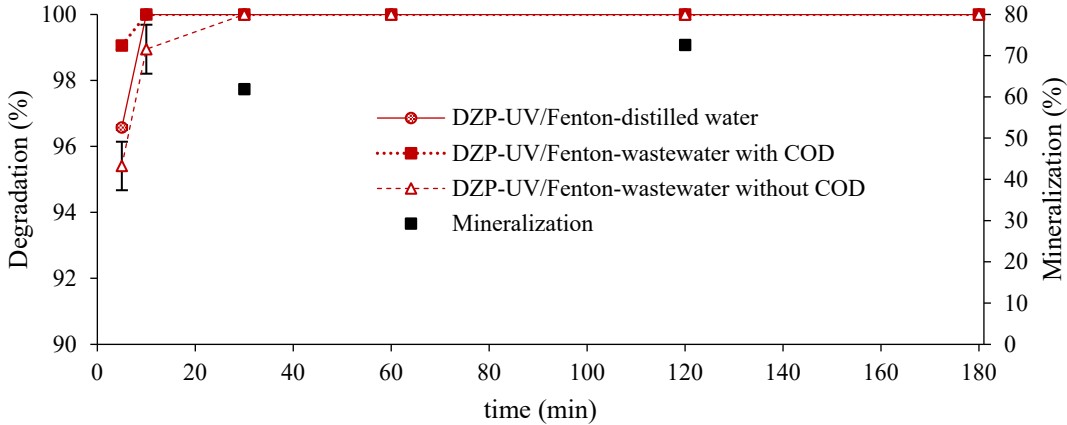

**Figure 5.** DZP degradation by photo-Fenton of technical water, wastewater without taking into account COD, and wastewater taking into account COD. [DZP] = 20 mg L$^{-1}$; pH 3.

The diazepam highest mineralization values, 61.9% and 72.6%, were reached at 30 and 120 min from the start of the reaction respectively, witnessing an increase of 14.6% between the intervals measured 90 min apart. This result is of great importance since although total degradation is obtained for DZP, there is a substantial increase in the per cent of mineralization at 120 min with respect to that achieved in technical water with the application of Fenton's reagent alone (12%), with a difference in the mineralization reached of 60.6% in both processes.

Taking in account the possible industrial application of this process in combination with other conventional treatments for wastewater, photo-Fenton is applicable before biological treatment. This is highly favorable since the microorganisms responsible for the remission of contaminants are not capable of eliminating these molecules. It is also important to note that the photo-Fenton process guarantees a satisfactory value of efficiency in the reduction of BOD$_5$ (59.8%) and COD (67.3%), that reduces the times of residence of the water in the system and increase the volume to be treated each day.

## 4. Conclusions

1.  The three processes of advanced oxidation, sonolysis, radiolysis, and photolysis combined with the Fenton reagent guarantee the total degradation of diazepam in synthetic matrices and more than 80% in a real matrix with partial mineralization.
2.  Diazepam sonolysis guarantees 28% degradation of the drug at 30.6 W and 862 kHz at 3 h of experimentation, while with photolysis 38% is achieved after 5 h. With the increase of the dose of irradiation increases the degradation of the drug for doses greater than 500 Gy, reaching the total degradation for a dose of 2500 Gy. In all cases, the process is favored in acidic conditions, with the best results for the ultrasonic and photolytic study at pH 2.5 and for the radiolytic at pH 3.

3. The combination of the processes with hydrogen peroxide guarantees the intensification of the same to the 10 min with an increase in more than 30%, 63%, and 21% for the sonolytic, radiolytic, and photochemical degradation, respectively.
4. Integration of the processes with Fenton reagent guarantees the total elimination of the drugs at 10 min in a synthetic matrix, and an increase in mineralization by more than 16%, 60%, and 12% for sonolytic, radiolytic degradation, and photochemistry, respectively.
5. All of the criteria for the evaluation of the energy consumption agree that the photo-Fenton process constitutes the one with the lowest energy consumption for the degradation of diazepam in the water matrix.
6. The degradation of the DZP in real residual water gave the best results in the experiments where the COD was taken into account to adjust the $H_2O_2$, and $Fe^{2+}$ concentrations. The photo-Fenton process guarantees total degradation using solar radiation as a source of energy after 10 min. A decrease in COD, and $BOD_5$ of waste water was achieved below the limits required by NC-27-2012 for classification B. The gamma-Fenton process guarantees maximum efficiency in decreasing COD, $BOD_5$, and TOC with 89.2%, 82.1%, and 88.1%, respectively.

**Author Contributions:** Conceptualization, U.J.J.H., M.M.A. and S.G.; methodology, U.J.J.H. and M.M.A.; experimental work, M.M.A., S.G.G. and M.A.G.M.; formal analysis, U.J.J.H. and M.M.A.; resources, U.J.J.H., M.M.A. and S.G.; data curation, M.M.A., S.G.G. and M.A.G.M.; writing—original draft preparation, U.J.J.H., M.M.A. and S.G.G.; writing—review and editing, U.J.J.H. and S.G.G.; visualization, M.M.A.; supervision, U.J.J.H. and S.G.G.; project administration, U.J.J.H.; funding acquisition, U.J.J.H. and S.G.G. All authors have read and agreed to the published version of the manuscript.

**Funding:** This work was supported by the projects TATARCOP, InSTEC, University of Havana, and CAPES-Dolé: CAPES-2020, Guadeloupe.

**Institutional Review Board Statement:** Not applicable.

**Informed Consent Statement:** Informed consent was obtained from all subjects involved in this study.

**Data Availability Statement:** The data presented in this study are available on request from the corresponding author.

**Conflicts of Interest:** The authors declare no conflict of interest.

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
