# Peer review of "Degradation of Diazepam with Gamma Radiation, High Frequency Ultrasound and UV Radiation Intensified with H2O2 and Fenton Reagent"

_processes, doi:10.3390/pr10071263_

Round 1

Reviewer 1 Report

Article Degradation of diazepam with gamma radiation, high frequency ultrasound and UV radiation intensified with H2O 2 and Fenton reagent

The article investigates the degradation of diazepam using gamma radiation, high frequency ultrasound, and UV radiation (artificial-solar), as well with each process intensified with oxidizing agents (H2O2 and Fenton reagent).

The topic of the work is relevant and within the scope of the journal.

However, it has several errors that make it considered that the article is not to be published.

First, I will detail the major comments:

- In the abstract it is said that the study is carried out on wastewater samples when in the article they use distilled water as a matrix. In the experimental part, it is not explained how the wastewater samples were extracted or from which plant it was used. In the conclusions, the BOD5 values ​​are emphasized, however, BOD5 analysis or any form of determination is not included. Therefore, the type of matrix used and the object of study of the work must be clearly defined.

- There are bibliographic citations that are not correct. For example ref 21. This article does not mention the process of radical recombination. Something similar with reference 38.

- There are wrong defined equations. For example 23.

minor comments

- In the experimental part, the irradiated area must be mentioned in the case of the use of the UV lamp in order to consider the power. The nominal power of the lamp of 11 watts is indicated.

- It should be clear if the experiments (gamma radiation, high frequency ultrasound, and UV radiation) are performed simultaneously or not

- Figure 1 should be separated into two images. One corresponding to the experiment with ultrasound and the other with radiation.

-In table 1, fields appear together that do not allow us to identify the values ​​obtained. It is also not clear why some have a standard deviation and others do not.

- There are text errors (misspelled words, blank spaces)

Author Response

Reviewer 1:

- In the abstract it is said that the study is carried out on wastewater samples when in the article they use distilled water as a matrix. In the experimental part, it is not explained how the wastewater samples were extracted or from which plant it was used. In the conclusions, the BOD5 values ​​are emphasized, however, BOD5 analysis or any form of determination is not included. Therefore, the type of matrix used and the object of study of the work must be clearly defined: A new section was included at the end of the manuscript. There, the results with a real wastewater are shown.

- There are bibliographic citations that are not correct. For example ref 21. This article does not mention the process of radical recombination. Something similar with reference 38: The references was changed by :

. 21: Wang, J., Chu, L. (2016) Irradiation treatment of pharmaceutical and personal care products (PPCPs) in water and wastewater: an overview. Radiation Physics and Chemistry. 2016, 125: p. 56–64. https://doi.org/10.1016/ j.radphyschem.2016.03.012.

. 38: Rossi-Bautitz, I., Pupo-Nogueira, R., Photodegradation of lincomycin and diazepam in sewage treatment plant effluent by photo-Fenton process. Catalysis Today, 2010. 151: 94-99.

- There are wrong defined equations. For example, 23: In the original sent manuscript, equations 22 and 23 are correct. It seems there was some mistake when transcribing these equations to the template. The right equations were included in the new version of the manuscript.

 - In the experimental part, the irradiated area must be mentioned in the case of the use of the UV lamp in order to consider the power. The nominal power of the lamp of 11 watts is indicated: The direct irradiated area was of 78.5 cm2. It was included in the manuscript.

- It should be clear if the experiments (gamma radiation, high frequency ultrasound, and UV radiation) are performed simultaneously or not: The AOPs processes were studied separately. It was specified in the first paragraph of the section “Material and methods”.

- Figure 1 should be separated into two images. One corresponding to the experiment with ultrasound and the other with radiation: done

-In table 1, fields appear together that do not allow us to identify the values ​​obtained. It is also not clear why some have a standard deviation and others do not:  The separation lines were included in the table for better identification of values, as suggested by the referee. The standard deviation was included only for the run 9 of each optimization series because three replicates were done in this point for verifying the reproducibility of experiments. A sentence was included explaining this.

- There are text errors (misspelled words, blank spaces): The errors were corrected.

Reviewer 2 Report

The manuscript entitled “Degradation of diazepam with gamma radiation, high frequency ultrasound and UV radiation intensified with H2O2 and Fenton reagent” by Manduca Artiles and co-authors explains the efficacy of different methods of DZP degradation (augmented with H2O2 and Fenton). The results are quite clear, as well as the methods performed. I especially liked the last section regarding energy consumption of the different methods. I do not have further comments. See below for (mostly) grammar and style comments.

Notation: as there are no line numbers, for each manuscript section, I will number each paragraph with “P” (ie. P1, P2) and line number with “L”, and re-start paragraph numeration on the next section.

Specific comments:

Abstract

P1.L1. Change “Degradation study” for “A degradation study”.

P1.L2. Change “and radiation UV” for “and UV radiation”.

P1.L6. Change “radiation and sonification” for “radiation, and sonification”.

P1.L8. Change “it was achieved 28.3 % degradation at” for “, it was achieved a 28.3 % degradation after”.

P1.L9. Change “photolysis a 38.2 % degradation was obtained at 300 minutes” for “photolysis, a 38.2 % degradation was obtained after 300 minutes”.

P1.L13. What was that shorter time? Specify.

Introduction

P1.L1. Change “groundwater and wastewater” for “groundwater, and wastewater”.

P1.L2-3. Change “pesticide,” for “pesticides”.

P1.L3. Use a better term than “marked interest”.

P1.L4. Change “on their effects” for “of their effects”.

P1.L3-L4. Is this information really “scarce”? I do not think so.

P1.L5. Here, the inefficiency to do what?

P1.L7. Change “lakes and to” for “lakes, and to”.

P1.L8. Change “tranquilizing and anticonvulsive” for “tranquilizing, and anticonvulsive”.

P1.L9-10. Explain briefly those ecosystem effects of this drug.

P3.L1. Change “only studies” for “only, studies”.

P3.L2. Change “radiation and its combination” for “radiation, and its combination”.

P4.L1. Change “work we study” for “work, we study”.

P4.L2. Change “sonolysis and” for “sonolysis, and”.

P4.L4. Change “The use of oxidizing” for “We hypothesize that the use of oxidizing”.

Materials and Methods

P1.L2. Change “water at pH” for “water, at pH”.

P1.L3. Change “and 3 adjusted” for “and 3, adjusted”.

P1.L4. Change “purity) of the Merck.” for “purity; Merck).”.

P2.L4. Change “sulfate and 98 %” for “sulfate, and 98 %”.

P3.L1. Change “gamma rradiation” for “gamma iradiation”.

P3.L2. Change “2.5, 5.0” for “2.5, and 5.0”.

P4.L3. Change “levels were evaluated The degradation” for “levels, were evaluated. The degradation”.

P4.L4. Change “temperature of” for “temperature”.

P5.L4. Change “of 5 L volume” for “of a 5 L volume”.

P6.L1. Change “processes a Shimadzu” for “processes, a Shimadzu”.

P6.L3. Change “injector and a UV” for “injector, and a UV”.

P6.L5. Change “flow and a ratio” for “flow, and a ratio”.

P7.L2. Change “Shimadzu TOC-V CSN,” for “Shimadzu (TOC-V CSN),”.

P7.L6. Change “HPLC respectively.” for “HPLC, respectively.”.

P8.L2. Change “expressions” for “equations”.

Results and Discussion

Effect of operational conditions on the degradation of DZP by radiolysis, sonolysis and photolysis

P1.L4. Change “the 580 kHz the electric” for “the 580 kHz frequency, the electric”. Change “8.7, 21.8” for “8.7, and 21.8”.

P1.L5. Change “10.4, 30.6 Watt” for “10.4, and 30.6 Watt”.

P1.L6. Change “2500, 5000” for “2500, and 5000”. And also, why is necessary to repeat this information? This is already on the Methods section.

P1.L8. Change “500 Gy the” for “500 Gy, the”.

P1.L9. Change “Gy a 100 %” for “Gy, a 100 %”.

P2.L1. Change “case the amount” for “case, the amount”.

P3.L4. Change “100 % decreasing” for “100 %, decreasing”.

P4.L3. Change “powers cavitation” for “powers, cavitation”.

P4.L4. Change “radicals thus increasing” for “radicals, thus increasing”.

P5.L2. Change “Although degradation” for “Although this degradation”, if that is the case.

Initial effect of pH on degradation of DZP

P1.L2. Change “photolytic and” for “photolytic, and”.

P1.L4. Change “862 kHz and a” for “862 kHz, and a”.

P2.L2. Change “5, 7” for “5, and 7”.

P3.L6. Change “facilitating The process” for “facilitating the process”.

P4.L1. Change “For radiolysi increasing” for “For radiolysis, increasing”. Change “the percent of degradation” for “the degradation”.

P4.L2-3. This phrase is too convoluted.

P5.L1. Change “case UV radiation” for “case of UV radiation”.

P5.L2. Change “In previous publication, this studied” for “In a previous publication, this study”.

P5.L3. Which pH value? Specify it to the reader again.

Effect of hydrogen peroxide on the degradation of DZP by sonolysis, radiolysis and photolysis

P1.L1. Change “H2O2 an increase” for “H2O2, an increase”.

P1.L2. Change “which are” for “which is”.

P2.L3. Change “rate, equation” for “rate, as shown by equations”.

P3.L1-L4. This entire paragraph needs to be rewritten. Is not clear at all.

Effect of the Fenton reagent on the degradation of DZP by sonolysis, radiolysis and photolysis

P2.L3-4. Here and elsewhere there is some repetition of information on this manuscript.

P3.L2. Change “sonolysis and radiolysis” for “sonolysis, and radiolysis”.

P3.L3. Change “reagent a 100 %” for “reagent, a 100 %”.

P4.L3. Change “high % of” for “high percentage of”.

P4.L4. Add a comma (,) before “respectively”.

P6.L1. Change “the % of mineralization responds” for “the percentage of mineralization corresponds”.

P8.L1. Change “Fe2+and” for “Fe2+, and”.

P9.L3. This part: “a maximum higher than 22.5 %” is not clear.

Evaluation of energy efficiency in the degradation of DZP with AOPs

P1.L3. Change “photolysis and DZP sonolysis” for “photolysis, and DZP sonolysis”.

P4.L2. Change “carbon and TOCf the” for “carbon, and TOCf the”.

P5.L2. Change “sonolysis and their” for “sonolysis, and their”.

Conclusions

P1.L1. Change “radiolysis and photolysis” for “radiolysis, and photolysis”.

P2.L3. Delete “On the other hand”.

P3.L2. Change “63 % and 21 %” for “63 %, and 21 %”.

P3.L3. Change “radiolytic and photochemical” for “radiolytic, and photochemical”.

P4.L3. Change “60 % and 12 % for sonolytic, radiolytic degradation and photochemistry” for “60 %, and 12 % for sonolytic, radiolytic degradation, and photochemistry”.

P6.L6. Change “BOD5 and TOC” for “BOD5, and TOC”.

P6.L7. Change “82.1 % and 88.1 % respectively” for “82.1 %, and 88.1 %, respectively”.

Author Response

Reviewer 2

We thank the reviewer appreciation of our work and his/her valuables comments and corrections. All recommended corrections were done and included in the corrected manuscript.

Abstract

P1.L1. Change “Degradation study” for “A degradation study”: done

P1.L2. Change “and radiation UV” for “and UV radiation”: done

P1.L6. Change “radiation and sonification” for “radiation, and sonification”: done

P1.L8. Change “it was achieved 28.3 % degradation at” for “, it was achieved a 28.3 % degradation after”: done

P1.L9. Change “photolysis a 38.2 % degradation was obtained at 300 minutes” for “photolysis, a 38.2 % degradation was obtained after 300 minutes”: done

P1.L13. What was that shorter time? Specify: In advanced oxidation processes, the use of antioxidants, in optimal amounts and conditions, increases the amount of free radicals in the system and, consequently, increases the reaction rate and decreases the degradation time of contaminants. In the abstract, at the request of the reviewer, this explanation was added with the following sentence: "…thanks to the increase in the amount of free radicals in water"

Introduction

P1.L1. Change “groundwater and wastewater” for “groundwater, and wastewater”: done

P1.L2-3. Change “pesticide,” for “pesticides”: done

P1.L3. Use a better term than “marked interest”: the sentence was removed

P1.L4. Change “on their effects” for “of their effects”: the sentence was removed

P1.L3-L4. Is this information really “scarce”? I do not think so: the sentence was removed

P1.L5. Here, the inefficiency to do what? To eliminate persistent pollutants. This was specified in the text.

P1.L7. Change “lakes and to” for “lakes, and to”: done

P1.L8. Change “tranquilizing and anticonvulsive” for “tranquilizing, and anticonvulsive”: done

P1.L9-10. Explain briefly those ecosystem effects of this drug: done, the next explanation was included: “Due to benzodiazepines interaction with the GABAA receptor, they may affect the func-tion of the nervous system of non-target species, such as aquatic organisms [13]. On the other hand, Subedi et al. showed that zebrafish (Danio rerio) larvae exposed to the mix-tures of psychotic drug residues, including the benzodiazepines, had affected immune system and gene expression [14].”

P3.L1. Change “only studies” for “only, studies”: the sentence was modified.

P3.L2. Change “radiation and its combination” for “radiation, and its combination”: done

P4.L1. Change “work we study” for “work, we study”: done

P4.L2. Change “sonolysis and” for “sonolysis, and”: done

P4.L4. Change “The use of oxidizing” for “We hypothesize that the use of oxidizing”: done

 Materials and Methods

P1.L2. Change “water at pH” for “water, at pH”: done

P1.L3. Change “and 3 adjusted” for “and 3, adjusted”: done

P1.L4. Change “purity) of the Merck.” for “purity; Merck).”: done

P2.L4. Change “sulfate and 98 %” for “sulfate, and 98 %”: done

P3.L1. Change “gamma rradiation” for “gamma iradiation”: done

P3.L2. Change “2.5, 5.0” for “2.5, and 5.0”: done

P4.L3. Change “levels were evaluated The degradation” for “levels, were evaluated. The degradation”: done

P4.L4. Change “temperature of” for “temperature”: done

P5.L4. Change “of 5 L volume” for “of a 5 L volume”: done

P6.L1. Change “processes a Shimadzu” for “processes, a Shimadzu”: done

P6.L3. Change “injector and a UV” for “injector, and a UV”: done

P6.L5. Change “flow and a ratio” for “flow, and a ratio”: done

P7.L2. Change “Shimadzu TOC-V CSN,” for “Shimadzu (TOC-V CSN),”: done

P7.L6. Change “HPLC respectively.” for “HPLC, respectively.”: done

P8.L2. Change “expressions” for “equations”: done

Results and Discussion

 Effect of operational conditions on the degradation of DZP by radiolysis, sonolysis and photolysis

P1.L4. Change “the 580 kHz the electric” for “the 580 kHz frequency, the electric”. Change “8.7, 21.8” for “8.7, and 21.8”: done

P1.L5. Change “10.4, 30.6 Watt” for “10.4, and 30.6 Watt”: done

P1.L6. Change “2500, 5000” for “2500, and 5000”. And also, why is necessary to repeat this information? This is already on the Methods section: The information was eliminated in this paragraph.

P1.L8. Change “500 Gy the” for “500 Gy, the”: The information was eliminated in this paragraph.

P1.L9. Change “Gy a 100 %” for “Gy, a 100 %”: done

P2.L1. Change “case the amount” for “case, the amount”: done

P3.L4. Change “100 % decreasing” for “100 %, decreasing”: done

P4.L3. Change “powers cavitation” for “powers, cavitation”: done

P4.L4. Change “radicals thus increasing” for “radicals, thus increasing”: done

P5.L2. Change “Although degradation” for “Although this degradation”, if that is the case: done

Initial effect of pH on degradation of DZP

P1.L2. Change “photolytic and” for “photolytic, and”: done

P1.L4. Change “862 kHz and a” for “862 kHz, and a”: done

P2.L2. Change “5, 7” for “5, and 7”: done

P3.L6. Change “facilitating The process” for “facilitating the process”: done

P4.L1. Change “For radiolysi increasing” for “For radiolysis, increasing”. Change “the percent of degradation” for “the degradation”: done

P4.L2-3. This phrase is too convoluted: The phrase was eliminated.

P5.L1. Change “case UV radiation” for “case of UV radiation”: done

P5.L2. Change “In a previous publication, this studied” for “In a previous publication, this study”:

P5.L3. Which pH value? Specify it to the reader again: done

Effect of hydrogen peroxide on the degradation of DZP by sonolysis, radiolysis and photolysis

P1.L1. Change “H2O2 an increase” for “H2O2, an increase”: done

P1.L2. Change “which are” for “which is”: done

P2.L3. Change “rate, equation” for “rate, as shown by equations”: done

P3.L1-L4. This entire paragraph needs to be rewritten. Is not clear at all: The paragraph was modified.  

Effect of the Fenton reagent on the degradation of DZP by sonolysis, radiolysis and photolysis

P2.L3-4. Here and elsewhere there is some repetition of information on this manuscript: the repeated information was removed.

P3.L2. Change “sonolysis and radiolysis” for “sonolysis, and radiolysis”: done

P3.L3. Change “reagent a 100 %” for “reagent, a 100 %”: done

P4.L3. Change “high % of” for “high percentage of”: done

P4.L4. Add a comma (,) before “respectively”: done

P6.L1. Change “the % of mineralization responds” for “the percentage of mineralization corresponds”: done

P8.L1. Change “Fe2+and” for “Fe2+, and”: done

P9.L3. This part: “a maximum higher than 22.5 %” is not clear: For the color scale of the Figure 3 it can be seen this value.

 Evaluation of energy efficiency in the degradation of DZP with AOPs

P1.L3. Change “photolysis and DZP sonolysis” for “photolysis, and DZP sonolysis”: done

P4.L2. Change “carbon and TOCf the” for “carbon, and TOCf the”: done

P5.L2. Change “sonolysis and their” for “sonolysis, and their”: done

Conclusions

P1.L1. Change “radiolysis and photolysis” for “radiolysis, and photolysis”: done

P2.L3. Delete “On the other hand”: done

P3.L2. Change “63 % and 21 %” for “63 %, and 21 %”: done

P3.L3. Change “radiolytic and photochemical” for “radiolytic, and photochemical”: done

P4.L3. Change “60 % and 12 % for sonolytic, radiolytic degradation and photochemistry” for “60 %, and 12 % for sonolytic, radiolytic degradation, and photochemistry”: done

P6.L6. Change “BOD5 and TOC” for “BOD5, and TOC”: done

P6.L7. Change “82.1 % and 88.1 % respectively” for “82.1 %, and 88.1 %, respectively”: done

Round 2

Reviewer 1 Report

An improvement in the article is observed.